# The Effect of Ramadan Intermittent Fasting on Food Intake, Anthropometric Indices, and Metabolic Markers among Premenopausal and Postmenopausal Women: A Cross-Sectional Study

**DOI:** 10.3390/medicina59071191

**Published:** 2023-06-24

**Authors:** Nada A. AlZunaidy, Abdulrahman S. Al-Khalifa, Maha H. Alhussain, Mohammed A. Mohammed, Hani A. Alfheeaid, Sami A. Althwab, MoezAlIslam E. Faris

**Affiliations:** 1Department of Food Science and Human Nutrition, College of Agriculture and Veterinary Medicine, Qassim University, Buraydah 51452, Saudi Arabia; h.alfheeaid@qu.edu.sa (H.A.A.); thaoab@qu.edu.sa (S.A.A.); 2Department of Food Science and Nutrition, College of Food and Agricultural Sciences, King Saud University, Riyadh 11451, Saudi Arabia; akhalifa@ksu.edu.sa (A.S.A.-K.); mhussien@ksu.edu.sa (M.H.A.); muhammed-awad@hotmail.com (M.A.M.); 3Department of Clinical Nutrition and Dietetics, College of Health Sciences, University of Sharjah, Sharjah 27272, United Arab Emirates

**Keywords:** nutrition, intermittent fasting, caloric restriction, time-restricted eating, menopausal, women’s health, dietary intake

## Abstract

*Background and objectives:* Ramadan intermittent fasting (RIF) is a monthlong practice in which Muslims fast during the whole day from sunrise to sunset. During this month, fasting people change their dietary behavior and alter their eating hours from day to night. The objective of the current study was to examine the effect of RIF on dietary consumption, anthropometric indices, and metabolic markers in healthy premenopausal (PRE-M) and postmenopausal (POST-M) Saudi women. *Materials and Methods:* The study included 62 women (31 PRE-M, 21–42 years, and 31 POST-M, 43–68 years). A structured questionnaire was used to collect socioeconomic data. Physical activity, anthropometric, dietary, and biochemical assessments were assessed before and at the end of the third week of Ramadan. *Results:* Socioeconomic data varied among participants. For both groups, observing RIF was associated with significantly (at either *p* ≤ 0.01 or *p* ≤ 0.05) lower intake of calories, macronutrients, minerals (excluding Na), and vitamins than before RIF. For the PRE-M group, the percentage of overweight participants decreased significantly (*p* ≤ 0.01) during Ramadan, while the percentage of obese participants remained unchanged. In contrast, for the POST-M group, the percentage of overweight participants increased significantly (*p* ≤ 0.05) during Ramadan, but dropped at the end of Ramadan. Following RIF, waist-to-hip ratio, body fat, and fat mass (FM) decreased in both groups. High-density lipoprotein cholesterol (HDL-C), fasting blood glucose (FBG), triglycerides (TG), and blood pressure (SBP, DBP) were generally maintained at acceptable normal levels in most participants before and at the end of RIF. However, low-density lipoprotein cholesterol (LDL-C) at the end of RIF was significantly lower than before, particularly in POST-M women. Age, occupation, and monthly income were the most important predictors for the changes in nutritional status and body fat upon RIF. *Conclusions*: observing RIF by PRE-M and POST-M Saudi women was associated with significant improvements in variable health indicators, with a few exceptions, and may help lower risk factors for chronic diseases, particularly among POST-M women. However, further studies with a larger sample size are required to determine and confirm the exact effect of RIF on these groups.

## 1. Introduction

Religious fasting has been observed by humans for thousands of years, with variable health implications reported in the literature [1]. According to the Islamic calendar, Ramadan is the ninth lunar month, during which healthy adult Muslims are mandated to fast from dawn to sunset, abstaining from all food and water, with the common practice being to have a large meal after dusk and a small meal before dawn [2], with no dietary restrictions between dusk and dawn [3]. During this month, the length of the daily fast varies according to the geographical area and time of year, ranging from 11–22 h according to the geographical location and season [2]. Fasting has been shown to elicit significant changes in dietary habits and food consumption patterns when combined with many lifestyle changes in physical activity [4], sleep patterns, and circadian rhythmic changes [5,6], leading to significant changes in anthropometric, cardiometabolic, glucoregulatory, and inflammatory parameters [7,8]. However, researchers have discovered that a dietary pattern specific to Ramadan contains unique properties that have not yet been identified as a dietary pattern model [7]. According to Daradkeh et al. [9], during Ramadan, Muslims’ dietary pattern change; they eat only two main meals—*sahur* before dawn and *iftar* at sunset—and dishes consumed during Ramadan are high in sugar, starch, protein, and fat. Notwithstanding an increase in fat intake, daily energy consumption did not decrease. However, the participants lost weight and reduced their waist circumference [8]. 

There are no dietary restrictions on the amount or type of food consumed following the breaking of the daily fast. Food consumption during Ramadan has been linked to significant changes in dietary patterns and the amount and frequency of food items [10]. Some studies have reported lower energy intake when breakfasting [10,11]. In contrast, it was observed that energy consumption was higher during the night hours of fasting days than during regular days [11]. They suggest that this energy can be derived from eating more energy-dense foods such as sweets [8,12] and fats [8,13]. Such dietary changes are significant during this month, particularly among people predisposed to metabolic diseases associated with these changes, such as obesity, diabetes, and hypertension [8]. However, during RIF, any changes in body mass, nutrient intake, and nutritional status may be influenced by the individuals’ level of physical activity [14]. Ramadan’s effects on energy balance and weight regulation have been thoroughly researched, but the results have been mixed [8]. Consequently, while some studies show weight loss (including lean mass and fat) [15,16], body weight and body composition remain stable [17,18] or even increase [19]. Decreased resting metabolic rate and physical activity may occur [20]. However, this appears to be offset by reduced sleep time [5], resulting in no effect on total daily energy expenditure measured by doubly labeled water [20]. As a result of RIF, body mass index (BMI) may or may not decrease, according to Mazidi et al. [20]. They linked BMI variations to the quality and quantity of foods consumed by Muslims in various countries and subcultures. RIF has been shown to reduce weight, BMI, body fat, and biochemical parameters such as triglyceride (TG) and fasting blood glucose (FBG). According to Jahrami et al. [8], body weight changes caused by Ramadan fasting are generally reversed after Ramadan and eventually return to the pre-Ramadan state. However, there is a relatively small weight variation throughout Ramadan. Further, Ramadan offers a chance to lose weight, but systematic and persistent lifestyle changes are required to accomplish long-term weight loss [17]. In addition, Jahrami et al. [8] noted that there were heterogeneous variations in body weight as a result of Ramadan fasting. They argue that dietary restrictions or energy intake limitations alone cannot account for these variations and that a variety of factors, such as dehydration, dietary changes, physical activity, and even sleeping patterns, may play a role. Conversely, low-density lipoprotein cholesterol (LDL-C) and fasting insulin increased [21]. There have been conflicting reports on the effect of RIF on blood hematological and biochemical parameters. Fasting blood sugar (FBG) and high-density lipoprotein cholesterol (HDL-C) levels decreased significantly, while TG, total cholesterol (TC), and LDL-C levels increased, according to Jahrami et al. [7]. 

Rathnayake et al. [22] found a substantial inverse relationship between middle-aged women’s quality of life (QoL) and menopausal symptoms. Despite no differences in calories, protein, fat, and various minerals and vitamin intake between premenopausal (PRE-M) and postmenopausal (POS-M) women, POS-M women had a significantly greater waist-hip ratio than PRE-M women [23]. Menopause is well known to cause an increase in body weight and fat tissue [24]. This suggests that the loss of estrogen after menopause affects body weight, partly by increasing visceral fat [25]. Bhurosy et al. [26] reported that POST-M women’s diets were of a higher quality than PRE-M women’s diets, despite their higher mean BMI, which may be attributed to physiological changes during menopause. Conversely, Nguyen et al. [27] found that POS-M women had lower nutrient intake levels (fat, protein, sugar, and some minerals) than PRE_M women. Moreover, Bashar et al. [28] used a menopausal-specific QoL questionnaire to conduct a cross-sectional study and reported that POST-M women had a higher risk of depression than PRE-M women. Further, they reported that those psychological symptoms, as well as “difficulty concentrating” and “fatigue,” are highly associated with depression scores and have a greater influence on QoL in early POST-M women than late POST-M women [28]. To the best of our knowledge, there is a paucity of studies examining the effect of RIF on nutritional, metabolic, and anthropometric markers among PRE-M and POST-M women. Therefore, the current study aimed to investigate the effects of observing RIF by PRE-M and POS-M women in Saudi Arabia on nutritional, anthropometric, and metabolic markers, and the associated socioeconomic factors.

## 2. Methods

### 2.1. Study Subjects

The research was done during Ramadan from the first of April to the first of May 2022. For those who were eligible, an information sheet was made available. The information was collected twice: once one week before Ramadan (R1) and once at the end of Ramadan’s third week, following 21–30 days of fasting (R2). Participants in Ramadan abstained from all food and drink (including water) from dawn till dusk (approximately 14 h). Participants in the study were not given any additional dietary recommendations and were asked to continue eating and exercising as usual before Ramadan. The sample size was determined according to the prevalence of satisfaction and the accuracy of the survey, and participants were chosen based on inclusion and exclusion criteria. Sixty-two women aged 21 to 68 were conveniently recruited using social media from Qassim district, Saudi Arabia. Before the study began, all participants provided their written, informed consent. Based on the questionnaire and the clinical information, the participants were split into two groups: PRE-M women (21–42 years old) and POST-M women (43–68 years old). Healthy women with regular menstrual cycles were classified clinically as PRE-M, whereas those whose menses had stopped at least a year before sample collection were classified as POST-M, both groups were then included. However, those with a history of smoking, CVD, pregnancy, cancer, hypertension, breastfeeding, as well as diabetes, and even non-diabetics but taking medications for sugar regulation, and those taking medications that lead to metabolic changes such as antiretroviral, corticosteroids, antiseizure, psychotropic, insulin, sulfonylurea, and thiazolidinediones [29], or experiencing a weight change of more than 3 kg right before the study period were excluded.

### 2.2. Socioeconomic Data Collection

The socioeconomic data was gathered using a structured questionnaire according to Ardawi et al. [30] and approved by a nutritional expert committee. Before beginning the study, the researcher conducted face-to-face interviews with the participants who had provided written, informed consent. The questionnaire collected participants’ demographic data, including questions about age, educational level, occupation level, income, and physical activity whether low (walking), medium (walking with exercise for a short period), or high (intensive exercise), which was validated by experts.

### 2.3. Dietary Intake and Analysis 

The daily food record method was employed. To make the best analysis of the dietary intake, the principal researcher and the assistants provided printed records for the participants with instructions, examples, and tables to record food for all days. The next step was to instruct each participant to record her food intake promptly in the provided table. Being excused from fasting according to Islamic Ramadan regulation, PRE-M women undergoing menstruation were requested to choose three days in *Sha’ban* (the month before Ramadan) and three days during Ramadan. The three days to be chosen are separated so that two are in the middle of the week and one is a weekend day. Participants were asked to record (in spoon size, number of cups, etc.) all foods and drinks, including snacks, appetizers, sauces, and seasonings, even for small portion sizes, and preparation methods. Extra sheets of food records were given to the participants when more space to record the foods eaten was required. The importance of accuracy and details was emphasized, and the researcher’s mobile phone number was written on each piece of paper in case the participant had any questions or inquiries. Participants were also asked to continue with their usual eating habits before and during Ramadan. The participants were given instructions and asked to accurately describe the meal’s type and amount using common measurements such as a tablespoon, teaspoon, cup, size of canned food by gram or liter, preparation method, and food additives used while cooking. To estimate dietary intakes of macro- and micronutrients, a nutrition analyzer software program designed for research and clinical use was used (ESHA Food Processor 11.9.13, 2020). Each type of food, fresh or cooked, was assigned a code using this software. The program then automatically analyzed the foods consumed and provided explicit quantities of nutrients such as calories, carbohydrates, protein, and so on for each participant based on daily intake for three dietary days. Nutritional breakdown information for each food item for up to 172 data fields, including water and macro- and micronutrients, are included in the ESHA master database, which consists of ingredients, recipes, and other food items. The nutrients obtained before and at the end of Ramadan were compared using a *t*-test.

### 2.4. Anthropometric Measurement

Anthropometric measurements, such as percent body fat (PBF), BMI, and fat mass (FM), were measured in duplicate using a multiple-frequency ACCUNIQ BC360 body composition analyzer (SELVAS Healthcare Inc., Daejeon, Korea) following the manufacturer’s instructions. Waist circumference (WC) and hip circumference (HC) were measured with a nonstretchable meter. The waist-to-height ratio (WHtR) was calculated by dividing WC by H, whereas the waist-to-hip ratio (WHR) was calculated by dividing WC by HC. BMI was calculated as a weight ratio in kg to square meter height (kg/m^2^). The BMI classification was done according to Romero-Corral et al. [31], the WHtR classification was done according to Ashwell et al. [32], and the WHR classification was done according to the WHO [33]. Moreover, based on methods in the body composition analyzer manual, body fat percent (BFP) and fat mass (FM) standard references were obtained.

### 2.5. Blood Samples Collection and Analysis

Before and at the end of Ramadan, blood samples were collected. Each subject had a venous blood sample (10 mL) drawn at the end of Ramadan for at least 8 h at both time points of the study. To eliminate the influence of timing and dietary intake on the measured biochemical parameters and to ensure uniform fasting duration at both times, samples were collected between 11 a.m. and 1 p.m. at both time points. The blood was divided into two aliquots. Within an hour of blood collection, the aliquot was centrifuged at 3000 rpm for 15 min, and the serum was coded and stored at −80 °C until used for biochemical analysis. In the laboratories of King Saud Hospital in Unaizah (Qassim district), participants’ blood parameters were analyzed using the XN1000-Cobas E411 (Sysmex-Rauch-Hitachi, Tokyo, Japan). Regarding the analyses, participants who were experiencing their menstrual period were asked to fast for 12 h before blood sample collection to ensure the consistency and validity of the results following all conditions for comparison before and at the end of Ramadan.

### 2.6. Blood Glucose and Lipid Profile Determination 

Colorimetric kits (Office, Shanghai, China) and a Dimension Xpand Plus Chemistry Analyzer were used to determine blood glucose levels and lipid profiles such as total cholesterol (TC), LDL-C, HDL-C, and triglycerides (TG). All measurements were taken following the manufacturer’s instructions.

### 2.7. Ethical Approval

The research protocol was approved by the regional research ethics committee in Qassim and registered at the national committee of Bio & Med, Ethics (NCBE) with registration numbers H-04-Q-001 and No. (1109570-1443). The work was carried out under the legal requirements and guidelines for good clinical practice.

### 2.8. Statistical Analysis

We calculated that 51 subjects would provide 80% power to detect a significant difference of 5% in lipid profile between prefasting and postfasting using a one-tailed paired-samples *t*-test with α = 0.05. We estimated a dropout rate of 10%. Thus, we planned to enroll a total of 57 participants from each group of women. The Statistical Package for Social Sciences (SPSS, version 2025) was used to perform the statistical analysis (SPSS Inc., Chicago, IL, USA). The study participants’ sociodemographic characteristics, anthropometric indices, and blood parameters were expressed as frequencies and percentages. A chi-squared test was used to determine whether or not there was a significant difference within the group before and at the end of the fasting month. At the two time points, the average intakes of energy and macro- and micronutrients were calculated as means. In terms of food intake and to determine whether or not there are significant differences between participants’ intake before and at the end of RIF, paired *t*-tests were used. Spearman correlation coefficients and simple regression analysis were used to determine the relationship between anthropometric indices and intake of calories, cholesterol, and vitamin D as dependent variables and socioeconomic characteristics and physical activity as independent variables.

## 3. Results

### 3.1. Socioeconomic Characteristics

The frequency distribution of the PRE-M and POST-M women in the study according to socioeconomic characteristics is shown in Table 1. The frequency distribution by age revealed that most of the PRE-M women (48.39%) were between 31 and 40 years, while most of the POST-M women (83.87%) were over 50. The percentage of PRE-M women with a university education was 64.51%, while for the POST-M women, this figure was 35.48%. Regarding employment, 35.48% of the PRE-M women worked for the government, while 83.87% of the POST-M women did not. About half of the PRE-M women (48.38%) had a monthly income of 5000 to 10,000 Saudi Riyal (SR), while over a third of the POST-M women (38.72%) had a monthly income of more than 15,000 SR (private sector). Both groups engaged in low physical activity before and during Ramadan, with 61.29% and 67.75% of the PRE-M women and 96.77% and 90.32% of the POST-M women engaged in low physical activity before and during Ramadan, respectively.

### 3.2. Dietary Intake 

Table 2 shows the average daily nutrient intake of the PRE-M and POST-M women in the study (daily food record, 3 days) before and at the end of Ramadan. A Student paired *t*-test was used to compare each constituent’s calculated average nutrient intake before and at the end of Ramadan for the PRE-M and POST-M groups. The average calorie intake of the PRE-M and POST-M women decreased significantly (*p* ≤ 0.01) from 1959.13 to 1823 kcal for PRE-M and 2111.98 to 1901.31 kcal for POST-M, at the end of RIF. Protein and carbohydrates for both the PRE-M and POST-M groups were not significantly different between the two periods, while dietary fiber was increased, with no significant difference between the two periods. At the end of fasting, total fat and cholesterol were significantly (*p* ≤ 0.05) lower, as was unsaturated fat (*p* ≤ 0.01). However, at the end of the fasting month, saturated fat in the PRE-M women was significantly (*p* ≤ 0.01) lower, whereas it was significantly (*p* ≤ 0.01) higher in the POST-M women. Most vitamin intakes were significantly lower at either *p* ≤ 0.01 or *p* ≤ 0.05 during fasting than during the corresponding nonfasting period for both groups. All major and trace mineral intakes were inadequate in both groups and were significantly lower (*p* ≤ 0.01) during fasting than the nonfasting period, except for sodium in the PRE-M group, which increased significantly (*p* ≤ 0.01) at the end of the Ramadan fasting month.

### 3.3. Anthropometric Indices of Participants 

Table 3 shows the BMI, WHtR, WHR, PBF, and FM for both the PRE-M and POST-M women. Table 3 shows a significant (*p* ≤ 0.01) difference in classification degree for both groups in terms of BMI, using the chi-squared test. Before fasting, the majority of the PRE-M women had a normal BMI (48.39%), followed by overweight (38.71%) and underweight (2%). In contrast, at the end of Ramadan, the percentage with normal BMI increased to 64.52%, the percentage of overweight PRE-M women decreased to 22.58%, and that of underweight PRE-M women increased to 6%. Furthermore, 48.39% of POST-M women were overweight before fasting, followed by those who were obese (32.26%), but at the end of Ramadan, the percentage of overweight POST-M women increased to 54.84%, and of obese POST-M women decreased to 22.58%. Before fasting, most PRE-M women (51.61%) had normal WHtR, followed by those who were underweight (25.81%); at the end of Ramadan, the percentage of POST-M women with normal WHtR did not change, while the percentage of underweight POST-M women increased to 32.26%. Furthermore, the percentage of overweight POST-M women decreased from 16.12% to 12.90% at the end of Ramadan, as did those who were extremely overweight. The percentage of obese POST-M women decreased from 41.94% to 22.58%, as did that of those who were overweight at the end of Ramadan. However, the percentage of overweight POST-M women increased from 25.81% to 48.39% at the end of the fast. Regarding WHR, most of the PRE-M women had a low ratio (90.32%), which increased significantly to 96.77% at the end of fasting, as did the percentage of POST-M women with low WHR, which increased from 41.94% to 58.07%. At the end of fasting, both groups’ normal and high WHR ratios decreased.

The percentage of PRE-M women with normal PBF increased from about 55% to about 65% at the end of fasting, as did those with high PBF. However, 45.16% of POST-M women had a very high PBF, which decreased to 41.94% at the end of fasting, whereas those with a high PBF increased from 32.26 to 35.48. The FM differed significantly (*p* ≤ 0.01) between participants. Before fasting, a majority of the PRE-M women had a normal rate of normal FM (51.61%), followed by a large minority with a high level (45.16%). At the end of Ramadan, the percentage of those with normal FM increased to 61.29% and those with a high level decreased to 35.48%. Before fasting, 87.10% of the POST-M women had high FM, which decreased to 83.87% at the end of Ramadan; while those with a normal level before fasting increased from 12.90% before Ramadan to 16.13% at the end of Ramadan.

### 3.4. Lipid Profile, Glucose, and Blood Pressure of the Study Participants

Table 4 shows the basic lipid profile parameters of the PRE-M and POST-M women before and at the end of Ramadan. Before fasting, most of the PRE-M women (96.8%) had low LDL-C, and all had low levels at the end of Ramadan. The percentage of POST-M women with low LDL-C decreased from 74.2% to 67.7%, while those with normal increased from 25.8 to 29.03%, from before to at the end of Ramadan, respectively. The PRE-M women had normal HDL-C levels before fasting, and only 3.2% had low levels at the end of Ramadan, whereas the POST-M women had normal levels before fasting, and 3.23% had high levels at the end of fasting. The percentage of PRE-M women with low cholesterol levels was found to be 80.65%, while the percentage with high levels was found to be 19.35%. However, the majority of the PRE-M women had either normal (77.4%) or high (22.6%) levels of cholesterol at the end of Ramadan. Before fasting, 61.3% of the POST-M women had high TC levels, which increased to 64.5% at the end of Ramadan at the expense of those with normal levels. Regarding TG, 96.8% of the PRE-M women had normal levels before and 90.33% at the end of Ramadan, whereas 93.6% of the POST-M women had normal levels before and 93.6% at the end of Ramadan. Before fasting, the percentage of PRE-M women with normal blood glucose levels was 83.9%, and at the end of Ramadan, this did not change. The percentage of POST-M women with normal levels was 41.9%, and those with high levels were 48.4%, and at the end of Ramadan, the percentage with normal levels increased to 64.5%, and the percentage with high levels decreased to 25.8%. Before fasting, the PRE-M women had either normal BP (77.4%), elevated (9.7%), or hypertension-I (12.9%), and at the end of Ramadan, the percentage with normal BP decreased to 74.2%, with elevated BP to 22.6%, and with hypertension-I to 3.23%. Before fasting, a significant minority of the POST-M women (45.2%) had hypertension-II, while 25.8% had hypertension-I; however, at the end of Ramadan, the percentage of POST-M women with hypertension-II decreased to 38.7% and the percentage with hypertension-I to 16.1%.

### 3.5. Factors Associated with the Nutritional Status of Participants

Table 5 displays the results of a simple linear regression analysis using socioeconomic characteristics and physical activity as independent variables and BMI and BFP as dependent variables. The BMI and BFP were positively or inversely correlated with the independent variables. According to simple regression analysis, a PRE-M woman’s occupation was inversely and significantly (*p* ≤ 0.05) associated with BMI but positively (*p* ≤ 0.05) associated with monthly income before fasting. However, at the end of the fast, physical activity was inversely and significantly associated with BMI (*p* ≤ 0.05). The results for POST-M women showed a positive association between age and BMI before and at the end of Ramadan; however, most other independent variables were inversely associated with BMI but at a nonsignificant rate. The occupation of PRE-M women was inversely and significantly associated (*p* ≤ 0.05) with BF before fasting, but the association was not significant at the end of Ramadan. The mean BF percent (BFP) of POST-M women was positively associated with age and monthly income but inversely with physical activity before and at the end of Ramadan.

Table 6 summarizes a simple linear regression analysis of the PRE-M and POST-M women’s socioeconomic characteristics and physical activity as independent variables, and calorie intake, cholesterol, and vitamin D levels as dependent variables before (R1) and at the end (R2) of Ramadan. The education level, monthly income, and physical activity of PRE-M women were inversely and significantly associated (*p* ≤ 0.01, *p* ≤ 0.05) with calorie intake before fasting, but calorie intake was inversely and significantly associated (*p* ≤ 0.05) with age, educational level, monthly income, and physical activity at the end of fasting. 

However, for the POST-M women, before fasting, calorie intake was inversely and significantly (*p* ≤ 0.05) associated with educational level and physical activity, and at the end of fasting was inversely and significantly (*p* ≤ 0.05) associated with educational level and positively with monthly income. 

The PRE-M women’s dietary intake of cholesterol before fasting was inversely and significantly associated (*p* ≤ 0.05) with physical activity, and at the end of fasting was inversely and significantly associated (*p* ≤ 0.05) with age and physical activity. Before fasting, the POST-M women’s dietary cholesterol intake was inversely and significantly associated with physical activity (*p* ≤ 0.05), but at the end of Ramadan, the association was not significant. The dietary intake of vitamin D of PRE-M women before and at the end of Ramadan was positively and significantly associated (*p* ≤ 0.05) with educational level and inversely and significantly associated (*p* ≤ 0.05) with age at the end of fasting. However, the POST-M women’s vitamin D intake was positively and significantly associated (*p* ≤ 0.05) with physical activity before and at the end of Ramadan.

## 4. Discussion

A cross-sectional survey was performed in the current study to examine the influence of observing RIF on food intake, anthropometric indicators, body composition, and the associated sociodemographic factors of Saudi PRE-M and POST-M women. In this study, the dietary intakes of PRE-M and POST-M adult women during Ramadan were compared using a Student paired *t*-test. For both groups, observing RIF was associated with a lower intake of calories, macronutrients, minerals (except Na), and vitamins than before RIF. This dietary pattern provided a more precise estimate of the difference between female dietary intake and what is required to improve our understanding of the impact of religious fasting on dietary behaviors and nutrient intake. The study observed that fasting during Ramadan was associated with significant changes in the intake of many food groups for both groups, implying that Ramadan has its own food intake and dietary pattern compared to the rest of the year. Participants’ energy intake before and at the end of Ramadan was influenced by their protein, carbohydrates, and total fat diet. Additionally, the amount of minerals and vitamins consumed is correlated with the consumption of fruits and vegetables.

Despite an increase in traditional Arabic sweets, cakes, pastries, and sugar-sweetened beverages unique to this month of the year, calorie intake during Ramadan was reduced for both PRE-M and POST-M women. The results of this study also revealed a clear trend toward a decreased intake of carbohydrates as sugar-containing foods by participants during the night hours of Ramadan, which is inconsistent with recent reports [34,35,36], indicating increased consumption of sugar-sweetened beverages after breaking the fast. Such high consumption of these foods can be linked to the traditions and customs common during this month, during which some sweets are produced, sold, and consumed. Some studies agree with the present study and found decreased energy intake [8,9], but another study by Ibrahim et al. [12] found increased energy consumption during Ramadan. This was mostly due to a higher intake of energy-rich nutrients, especially sweets [7,11] and fats [7,12]. Despite increased sugar consumption during Ramadan, especially among diabetic patients, fasting and associated higher sugar intake had no negative glucometabolic effects on healthy subjects [37]. Still, the effects on diabetic patients remain unknown. 

The present study found a decrease in protein intake by Saudi women, which is inconsistent with the reported increase in protein food consumption during Ramadan [11,38]. Whether the reported higher protein intake during Ramadan relates to its satiating effect is debatable. A meta-analysis of the effects of increased protein intake on satiety found that high-protein meals increased satiety rates more than low-protein meals [39]. As a result, this study’s total energy and macronutrient intake before and during Ramadan is a hallmark of intermittent fasting [40] and maybe a more practical dietary modification than calorie restriction [41]. In the latter study by Shatila et al. [41] intakes of cereals, cereal-based products, pasta, eggs, nuts and seeds, milk and dairy, and fats and oils were lower during Ramadan compared to the rest of the year (*p* < 0.05), whereas intakes of vegetables, dried fruit, Arabic sweets, cakes and pastries, and sugar-sweetened beverages were higher (*p* < 0.05). Intakes of nutrients, including carbohydrates, cholesterol, calcium, beta-carotene, vitamin C, folate, and magnesium, reflected these dietary group differences.

At the same time, salads, dates, and dried apricots are important traditional and cultural foods of Ramadan meals [41], and, unfortunately, we observed that the intake of such foods was low. In this study, low vegetable and fruit intake and a low intake of fiber-rich foods at night during Ramadan could explain the lower dietary fiber, total vitamins, and total minerals during Ramadan. The methodological and cultural differences in the study population can be attributed to discrepancies in the study results regarding the effect of Ramadan on dietary intake [42]. Variation in nutritional assessment tools is a major factor in conflicting results across studies examining food and dietary intake changes. Another major reason for the disparities in studies of Ramadan eating is the unique cultural and traditional food behaviors adopted by different populations of different ethnic and cultural backgrounds, all while adhering to the rules of the *halal* food system allowed for Muslims [42]. This difference between the PRE-M and POST-M women could be attributed to the fact that the transition to menopause significantly impacts cardiovascular health and response to various changes in diet and lifestyle [43]. 

In terms of BMI and WHtR, the results showed that for the PRE-M group, the percentage of overweight participants decreased significantly while the percentage of obese participants remained unchanged, whereas, for the POST-M group, the percentage of overweight participants increased significantly while the percentage of obese participants decreased at the end of Ramadan. After RIF, both groups’ body fat % (BFP) and fat mass (FM) decreased. The decrease in energy and macronutrient intake during Ramadan indicated that fasting during Ramadan significantly decreased BMI, BF, and FM, which was consistent with the findings of Mazidi et al. [20], who studied changes in cardiometabolic risk factors and anthropometric parameters during Ramadan. Nachvak et al. [44] also reported that fasting during Ramadan positively impacts weight, BMI, body fat, and some biochemical parameters. Furthermore, the negative effect of Ramadan fasting on body fat and BMI is expected to result in a significant reduction in waist circumference, observed in both groups during Ramadan. Although both groups’ body fat levels were significantly lower at the end of Ramadan, so was their waist circumference. This may enable women to develop a pattern of body fat redistribution following Ramadan. This result coincided with the BMI, indicating a stronger relationship between BMI and waist circumference. Although BMI, WC, and WHtR are all strongly correlated, WHtR is an independent predictor of early health risk after controlling for age, gender, and BMI [45]. According to this fact, at the end of Ramadan, the POST-M women had a higher health risk than the PRE-M women because women experience a variety of health issues during both the PRE-M and POST-M periods [43]. Both groups had low waist-to-hip ratios (WHR), but the POST-M women had a higher proportion of normal and high WHR. 

The findings supported previous research on reduced waist circumference throughout Ramadan [46,47,48]. In addition, a previous study found no significant changes in WHR in fasting women [49]. The PRE-M and POST-M groups both showed significantly decreased body fat percentage at the end of fasting. Throughout Ramadan, many studies reported decreased body fat percentage [21,50]. Despite the women’s reported decrease in body fat percentage during RIF, there were also conflicting results that showed an increase in body fat percentage during Ramadan [51]. However, in this study, both groups had a significant reduction in fat mass, which could be attributed to the fact that eating fat-rich foods as the primary energy source while fasting may play a role in lowering body fat percentage [51]. A systematic review and meta-analysis found that fasting during Ramadan resulted in statistically significant weight and body composition reductions, such as fat mass (kg) and lean mass, in people who were overweight or obese but not in those who were normal weight [51]. In general, Kozakowski et al. [52] reported that compared to PRE-M women, Post-M women have a higher prevalence of weight gain and a higher risk of many diseases related to obesity, and taking into account the growing frequency of overweight and obese women as a result of metabolic problems that appear in women during menopause should be considered as a socioeconomic factor.

The findings revealed that most participants’ HDL-C, FBG, TG, and blood pressure (SBP, DBP) levels were generally maintained at acceptable normal levels before and after RIF but LDL-C levels were significantly lower during RIF than before, particularly for the POST-M group. HDL-C was found to be normal at the end of Ramadan in both the PRE-M and POST-M women. However, a study has found an increase in HDL-C at the end of Ramadan [44], while another study has found a decrease [51]. These differences in HDL-C levels between the present study and previous ones could be attributed to the effect of Ramadan fasting on serum lipid levels, which is closely related to dietary habits and other lifestyle changes [51]. The levels of triglycerides (TG), total cholesterol (TC), fasting blood glucose (FBG), and blood pressure (SBP, DBP) were significantly decreased in both groups at the end of Ramadan. In agreement with the present study, a previous study found reduced cholesterol and triglyceride levels [53]. However, another study found no significant changes in cholesterol or triglyceride levels [50]. There is also evidence of increased cholesterol and triglycerides after RIF [54]. A study found no significant decrease in glucose levels in the blood [55]. Others reported higher [56] or lower [51] fasting blood sugar levels following RIF. Fasting during Ramadan has little impact on blood pressure or blood glucose levels, as reported by Bener et al. [56]. During intermittent fasting, lower blood pressure could be due to age, weight loss, or inhibition of catecholamine production, resulting in a reduction in sympathetic tone and, as a result, lower blood pressure, heart rate, and cardiac output [51]. According to the current research, both groups’ fundamental hematological indicators drastically decreased at the end of Ramadan. Differences in metabolic markers between this study and previous ones could explain differences in dietary habits, calorie intake, fasting days, daily fasting period, sampling time, genetic propensity, and daily activity.

According to this study, before fasting, the determinant factors associated with the BMI of PRE-M women were occupation and monthly income; women’s occupations were significantly associated with those with decreased BMI, while income was significantly associated with those who reported increased BMI. However, at the end of Ramadan, physical activity significantly decreased the BMI. Before fasting, women’s PBF was significantly lower, and no significant association with socioeconomic characteristics or physical activity was observed at the end of Ramadan. BMI was significantly increased before and at the end of Ramadan in POST-M women only with age, whereas PBF increased significantly with age and monthly income but decreased with physical activity. According to Alsaif et al. [57], obese women were likelier than non-obese women to be inactive. Improved interventions are required to encourage the adoption and long-term maintenance of physical activity, which can lead to better weight control, abdominal adiposity, and chronic disease risk [58]. This is primarily because physical activity among Saudi women is the primary factor in the high prevalence of obesity. Moreover, the rising monthly income levels lead to an increase in the number of overweight women, which means that a higher income leads to more disposable income that can be spent on higher-calorie foods, especially given the population’s lack of awareness of the health risks associated with obesity.

Before fasting, the PRE-M women’s calorie intake was associated with increased income. It decreased with education level and physical activity. However, at the end of Ramadan, it decreased with age, education level, and physical activity and increased with income. The caloric intake of the POST-M women decreased significantly with education level and physical activity before fasting and increased significantly with income at the end of Ramadan. Cholesterol levels were significantly lower with physical activity in both groups, but at the end of Ramadan, they were significantly lower with age and physical activity in the PRE-M women. Before fasting, vitamin D levels were significantly higher with education, but at the end of Ramadan, they increased with education but decreased with age.

According to this study, physical activity significantly increased vitamin D levels in POST-M women before and at the end of Ramadan. According to Fernandes et al. [59], exercising outside would provide both the benefits of physical activity and sun exposure, namely vitamin D synthesis, and it was discovered that increased plasma concentrations of vitamin D occur with physical activity both indoors and outdoors. The current findings show that low education and occupation can be contributed to dietary habits differences. Women with lower socioeconomic status ate less fish and vegetables and more meat, fried foods, table sugar, pasta, and potatoes [60]. 

Lower socioeconomic groups consumed less iron, calcium, vitamin A, and vitamin D. Education and occupation explained a dietary pattern that favored lower social class groups. Education and occupation effects were additive or synergistic for some foods and nutrients. Our findings suggest that both indicators should be evaluated to provide a complete picture of social inequalities in dietary habits.

The strength of our study can be explained by the fact that we looked at both PRE-M and POST-M women from the same community ensuring homogeneity and avoiding inequalities. Extensive questionnaires, physical measures, and laboratory tests were used to identify the factors influencing patients’ nutritional intake and health indices before and during RIF. Food consumption and biochemical and anthropometric markers were found to differ between the two groups; however, our data demonstrate that the observance of RIF exacerbates this. Furthermore, our findings imply that RIF has some health benefits in both groups. Nutrition research suffers from variations in participants’ dedication to the diet or intermittent fasting regimen, known as compliance or adherence. However, because our study was conducted in an Islamic country, and, as such, everyone adheres to the system completely from 4 a.m. *Fajr* (dawn) until 6 p.m. *Maghrib* (sunset) at an average of 15 h per day for a full month, which is uncommon in studies conducted in non-Islamic countries, where fasting systems can be broken or violated in any way. This means that the location of the current research and the quality of the time-bound participants in the intermittent fasting period are strengths of this study. 

The relatively small sample size was a limitation of this study because it focused on a specific Saudi Arabian region. It was not easy to cover the entire country with the study due to the large size of the country. Further, the results of the current work cannot be generalized over the rest of the year because of the short period of Ramadan. Moreover, all participants were drawn from the same community with an average income to ensure sample homogeneity and avoid disparities and confusion between cultural and socioeconomic factors. Because it is difficult to find the same number of nonfasting Ramadan participants who meet the same criteria as those who fasted, and because different countries have different physiological and dietary habits, the nonfasting control group was excluded. Instead, on fasting days, measurements were taken of women exempt from fasting due to menstruation. 

## 5. Conclusions

Participants consumed low calories, macronutrients, minerals, and vitamins during RIF. BMI, WC, waist-to-height ratio, BFP, and FM were significantly decreased at the end of Ramadan in both the PRE-M and POST-M groups. However, while LDL-C was significantly low, HDL-C levels were normal in PRE-M and POST-M women. At the end of Ramadan, participants’ TG, TC, FBG, and SBP/DBP levels were significantly decreased. According to the study, age, occupation, monthly income, and physical activity were the most important factors influencing women’s nutritional status and body composition. These findings imply that by encouraging exercise as well as nutritional guidance and support not only during the RIF period but also after it, the positive effects achieved during RIF can be enhanced, and weight regain after RIF and a fall in body composition standards can be avoided.

## Figures and Tables

**Table 1 medicina-59-01191-t001:** Frequency distribution of PRE-M (*n* = 31) and POST-M (*n* = 31) women according to socioeconomic characteristics and physical activity.

Variable	PRE-M (*n* = 31)				*p*-Value		POST-M (*n* = 31)			*p*-Value		
	*n*			%			*n*		%			
Age (year)												
21–30	12			38.71	0.044		-		-	0.0001		
31–40	15			48.39			-		-			
41–50	4			12.90			5		16.13			
>50	-			-			26		83.87			
Educational level												
Illiterate	-			-	0.001		1		3.23	0.0003		
Secondary	3			9.68			7		22.58			
Intermediate	1			3.23			1		3.23			
Diploma	4			12.90			10		32.25			
University	20			64.51			11		35.48			
Postgraduate	3			9.68			1		3.23			
Occupation												
Student	4			12.90	0.362		-		-	0.001		
Government employee	11			35.48			3		9.68			
Special job	8			25.81			2		6.45			
Not working	8			25.81			26		83.87			
Monthly income (SR)												
Less than 5000	1			3.23	0.005		2		6.45	0.041		
5000 to 10,000	15			48.38			7		22.58			
10,000 to 15,000	8			25.81			10		32.25			
More than 15,000	7			22.58			12		38.72			
*Physical activity level*	PRE-M (*n* = 31)						POST-M (*n* = 31)					
	R1			R2			R1			R2		
	*n*	%	*p*-Value	*n*	%	*p*-Value	*n*	%	*p*-Value	*n*	%	*p*-Value
Low	19	61.29	0.004	21	67.75	0.048	30	96.77	0.001	28	90.32	0.007
Moderate	5	16.13		10	32.25		-	-		-	-	
High	7	22.58		0	0		1	3.23		3	9.68	

The significance level was accepted according to chi-squared; F, frequency; R1, before Ramadan; R2, at the end of Ramadan. Physical activity: low (walking), medium (walking with exercise for a short period), or high (intensive exercise).

**Table 2 medicina-59-01191-t002:** Average daily intake of macro- and micronutrients (average of 3 days) before (R1) and at the end of Ramadan (R2) with the PRE-M and POST-M women.

Parameter	PRE-M (*n* = 31)		*t*-Test	*p*-Value	POST-M (*n* = 31)		*t*-Test	*p*-Value
	R1	R2			R1	R2		
Macronutrients								
Calories (kcal)	1959.13 ± 27.18	1823.34 ± 22.65	2.95 **	0.006	2111.98 ± 21.18	1901.31 ± 17.36	4.68 **	0.005
Protein (g)	77.87 ± 12.60	68.17 ± 16.55	1.81	0.081	83.47 ± 12.78	76.21 ± 11.32	1.92	0.065
Carbohydrates (g)	196.22 ± 24.34	186.22 ± 17.83	1.34	0.176	211.77 ± 22.80	190.74 ± 19.52	1.56	0.129
Dietary fiber (g)	7.53 ± 3.40	9.04 ± 5.51	−1.42	0.167	10.35 ± 4.91	11.06 ± 5.14	−0.73	0.473
Total fat (g)	93.53 ± 18.33	82.75 ± 13.32	1.98 *	0.050	94.56 ± 10.24	82.13 ± 9.06	2.29 *	0.029
Saturated fat (g)	26.55 ± 8.12	18.76 ± 6.91	7.59 **	0.003	29.87 ± 8.42	35.95 ± 9.86	−1.80 *	0.022
Unsaturated fat (g)	60.45 ± 11.15	48.99 ± 8.17	3.86 **	0.001	59.12 ± 14.78	43.24 ± 13.90	5.89 **	0.001
Cholesterol (mg)	260.81 ± 26.7	246.08 ± 19.2	0.356 *	0.024	336.44 ± 27.91	225.67 ± 15.50	1.84 *	0.016
Vitamins								
Vit A µg (RE)	238.96 ± 36.10	145.28 ± 15.40	1.37 *	0.018	394.49 ± 74.20	305.42 ± 74.10	0.47 *	0.041
Thiamin (mg)	0.54 ± 0.43	0.34 ± 0.18	2.93 **	0.006	0.68 ± 0.73	0.36 ± 0.54	2.71 *	0.011
Riboflavin(mg)	0.71 ± 0.48	0.40 ± 0.20	3.08 **	0.004	1.18 ± 0.50	0.63 ± 0.45	2.00 *	0.050
Niacin (mg)	12.27 ± 4.00	11.68 ± 3.7	0.302	0.765	15.42 ± 11.30	10.69 ± 8.70	2.18 *	0.037
Vit.B6 (mg)	0.61 ± 0.57	0.55 ± 0.55	0.711	0.482	0.90 ± 1.11	0.63 ± 0.88	1.37	0.180
Vit.B12 (mcg)	1.49 ± 1.30	1.01 ± 1.20	0.985	0.333	2.49 ± 0.53	2.45 ± 0.52	0.03	0.976
Vit..C (mcg)	23.00 ± 8.70	17.11 ± 6.10	1.04	0.308	32.40 ± 4.70	30.31 ± 3.21	0.32	0.754
Vit.E- (mg)	2.23 ± 2.40	1.10 ± 1.52	2.20 *	0.035	2.05 ± 1.51	0.96 ± 0.94	4.61 **	0.001
Vit.D–(mcg)	1.16 ± 1.20	0.51 ± 0.68	2.90 **	0.007	2.11 ± 0.98	0.65 ± 0.30	1.86 *	0.037
Vit. K (mcg)	24.87 ± 13.70	30.78 ± 16.80	−0.265	0.793	25.11 ± 3.81	11.41 ± 1.94	2.08 *	0.046
Folate (mcg)	143.34 ± 11.10	81.68 ± 7.71	2.87 **	0.007	181.80 ± 29.40	95.26 ± 19.91	1.63	0.113
Minerals								
Calcium (mg)	408.05 ± 23.61	366.84 ± 18.74	0.902 *	0.034	564.09 ± 31.90	388.45 ± 21.30	3.09 **	0.004
Copper (mg)	0.68 ± 0.64	0.49 ± 0.36	1.69	0.101	0.75 ± 0.19	0.74 ± 0.17	0.03	0.973
Magnesium (mg)	90.84 ± 8.81	79.23 ± 5.93	0.640	0.527	104.77 ± 55.90	80.49 ± 48.71	1.81 *	0.048
Iron (mg)	8.69 ± 4.70	7.76 ± 3.40	0.962	0.344	11.42 ± 7.10	9.22 ± 6.20	0.72	0.475
Phosphorus (mg)	326.99 ± 26.72	295.50 ± 19.10	0.480	0.635	383.19 ± 25.42	267.64 ± 17.33	2.95 **	0.006
Potassium (mg)	481.43 ± 86.50	322.14 ± 57.91	0.836 *	0.013	1459.59 ± 93.51	1366.69 ± 47.81	0.55	0.588
Selenium (mg)	40.79 ± 7.91	31.95 ± 3.73	1.03	0.311	41.22 ± 6.22	27.66 ± 2.80	2.59 *	0.014
Sodium (mg)	1834.06 ± 13.80	2482.38 ± 15.31	−1.81 *	0.018	2773.80 ± 14.83	2299.28 ± 19.51	0.44 *	0.016
Zinc (mg)	3.38 ± 2.93	2.62 ± 2.10	1.17	0.251	5.15 ± 3.73	3.66 ± 3.12	1.06	0.298

The significance level was accepted at * *p* ≤ 0.05; ** *p* ≤ 0.01 according to the *t*-test. R1, before Ramadan; R2, at the end of Ramadan.

**Table 3 medicina-59-01191-t003:** Anthropometric measures of PRE-M and POST-M women before (R1) and at the end of Ramadan (R2) fasting month.

Indices	PRE-M (*n* = 31)				*p*-Value	POST-M (*n* = 31)				*p*-Value
	R1		R2			R1		R2		
	*n*	%	*n*	%		*n*	%	*n*	%	
BMI (kg/m²)										
Underweight (<18.5) **	2	6.45	2	6.45	0.001	-	-	-	-	0.012
Normal (18.5–24.9)	15	48.39	20	64.52		6	19.35	7	22.58	
Overweight (25–29.9)	12	38.71	7	22.58		15	48.39	17	54.84	
Obesity (≥30)	2	6.45	2	6.45		10	32.26	7	22.58	
Total	31	100.0	31	100.0		31	100.0	31	100.0	
Waist-to-height ratio (WHtR)										
Underweight (≤0.41)	8	25.81	10	32.26	<0.001	2	6.45	2	6.45	0.005
Normal (0.42–0.48)	16	51.61	16	51.61		2	6.45	3	9.68	
Overweight (0.49–0.53)	5	16.12	4	12.90		8	25.81	15	48.39	
Very overweight (0.54–0.57)	1	3.23	-	-		6	19.35	4	12.90	
Obesity (≥0.58)	1	3.23	1	3.23		13	41.94	7	22.58	
Total	31	100.0	31	100.0		31	100.0	31	100.0	
Waist-to-hip ratio (WHR)										
Low (≤0.80)	28	90.32	30	96.77	<0.001	13	41.94	18	58.07	0.008
Normal (0.81–0.85)	2	6.45	1	3.23		9	29.03	7	22.58	
High (≥0.86)	1	3.23	-	-		9	29.03	6	19.35	
Total	31	100.0	31	100.0		31	100.0	31	100.0	
Body fat % (BFP, Kg)										
Low (<21)	5	16.12	5	16.12	<0.001	-	-	-	-	0.125
Normal (21–35.5)	17	54.85	20	64.52		7	22.58	7	22.58	
High (35.6–40)	6	19.35	3	9.68		10	32.26	11	35.48	
Very high (>40)	3	9.68	3	9.68		14	45.16	13	41.94	
Total	31	100.0	31	100.0		31	100.0	31	100.0	
Fat Mass (FM, kg)										
Low (<15)	1	3.23	1	3.23	0.001	-	-	-	-	<0.001
Normal (15–26.8)	16	51.61	19	61.29		4	12.90	5	16.13	
High (>26.8)	14	45.16	11	35.48		27	87.10	26	83.87	
Total	31	100.0	31	100.0		31	100.0	31	100.0	

The significance level was accepted according to chi-squared; F, frequency; R1, before Ramadan; R2, at the end of Ramadan. ()**, reference level.

**Table 4 medicina-59-01191-t004:** Frequency distribution of metabolic parameters (lipid profile and glucose) and blood pressure levels of PRE-M and POST-M women before (R1) and the end of Ramadan (R2) fasting month.

Parameter	PRE-M (*n* = 31)				*p*-Value	POST-M (*n* = 31)				*p*-Value
	R1		R2			R1		R2		
	*n*	%	*n*	%		*n*	%	*n*	%	
LDL-C (mmol/L)										
Low (<3.9) **	30	96.8	31	100.0	0.003	23	74.2	21	67.7	<0.001
Normal (3.9–4.9)	1	3.2	-	-		8	25.8	9	29.0	
High (>4.9)	-	-	-	-		-	-	1	3.2	
HDL-C (mmol/L)										
Low (<0.83)	-	-	1	3.2		1	3.28	-	-	<0.001
Normal (0.83–2.49)	31	100.0	30	96.8	<0.001	30	96.8	30	96.8	
High (>2.49)	-	-	-	-		-	-	1	3.3	
Total	31	100.0	31	100.0		31	100.0	31	100.0	
Total cholesterol (mmol/L)										
Low (<2)	25	80.6	-	-	0.017	-	-	2	6. 5	<0.001
Normal (2.0–5.2)	-	-	24	77.4		12	38.7	9	29.0	
High (>5.2)	6	19.4	7	22.6		19	613	20	64.5	
Triglycerides (mmol/L)										
Low (<0.34)	1	3.2	3	9. 7	<0.001	-	-	2	6. 5	<0.001
Normal (0.34–2.28)	30	96.8	28	90.3		31	100.0	29	93.6	
High (>2.28)	-	-	-	-		-	-	-	-	
Fasting blood glucose (mmol/L)										
Low (<3.9)	4	12.9	5	16.1	<0.001	3	9.7	3	9. 7	0.004
Normal (3.9–6.1)	26	83.9	26	83.9		13	41.9	20	64.5	
High (>6.1)	1	3.2	-	-		15	48.4	8	25.8	
Total	31	100.0	31	100.0		31	100.0	31	100.0	
Blood pressure (Systolic/Diastolic, mm Hg)										
Normal (120/80)	24	77.4	23	74.2	0.049	6	19.4	10	32.3	0.021
Elevated (120–129/<80)	3	9.7	7	22.6		3	9.7	4	12.9	
Hypertension-I (130–139/80–89)	4	12.9	1	3.2		8	25.8	5	16.1	
Hypertension-II (≥140/≥90)	-	-	-	-		14	45.2	12	38.7	

The significance level was accepted according to chi-squared; F, frequency; R1, before Ramadan; R2, during Ramadan. ()**, reference level.

**Table 5 medicina-59-01191-t005:** Simple linear regression analysis between socioeconomic, physical activity, and BMI and BFP before (R1) and at the end of Ramadan (R2) for PRE-M and POST-M women.

Dependent Variables/Independent Variables	PRE-M (*n* = 31)								POST-M (*n* = 31)							
	*BMI* (kg/m^2^)				*BFP*				*BMI* (kg/m^2^)				*BFP*			
	R1		R2		R1		R2		R1		R2		R1		R2	
	β	*p*-Value	β	*p*-Value	β	*p*-Value	β	*p*-Value	β	*p*-Value	β	*p*-Value	β	*p*-Value	β	*p*-Value
Socioeconomic factors																
Age	0.03	0.990	0.15	0.558	0.04	0.641	0.06	0.952	0.14 **	0.007	0.13 *	0.024	0.17 **	0.009	0.29 *	0.017
Education level	−0.10	0.781	−0.27	0.529	−0.08	0.591	−0.16	0.310	−0.09	0.755	−0.17	0.482	−0.24	0.166	−0.22	0.153
Occupation	−0.31 *	0.038	−0.26	0.464	−0.22 *	0.050	−0.019	0.894	−0.02	0.898	−0.02	0.901	−0.03	0.973	−0.02	0.837
Monthly income	0.18 *	0.042	0.12	0.648	0.13	0.193	0.16	0.127	0.21	0.276	0.01	0.972	0.33 **	0.007	0.29 **	0.001
Physical activity																
High	−0.16	0.564	−0.37	0.192	−0.02	0.817	−0.09	0.438	−0.06	0.820	−0.02	0.914	−0.12	0.465	−0.11	0.435
Moderate	−0.46	0.089	−0.62 *	0.030	−0.05	0.629	−0.08	0.477	−0.05	0.904	−0.03	0.920	−0.38	0.112	−0.37 *	0.050
Low	−0.16	0.149	−0.21 *	0.031	−0.04	0.588	−0.03	0.661	−0.88	0.140	−0.69	0.150	−0.57*	0.018	−0.48	0.125

The significance level was accepted at * *p* ≤ 0.05; ** *p* ≤ 0.01.

**Table 6 medicina-59-01191-t006:** Simple linear regression analysis between socioeconomic, physical activity, and dietary intake of calories, cholesterol, and vitamin D before (R1) and at the end of Ramadan (R2) for PRE-M and POST-M women.

Dependent Variables/Independent Variables	PRE-M (*n* = 31)											
	Calories				Dietary Cholesterol				Vitamin D			
	R1		R2		R1		R2		R1		R2	
	β	*p*-Value	β	*p*-Value	β	*p*-Value	β	*p*-Value	β	*p*-Value	β	*p*-Value
Socioeconomic factors												
Age	−0.03	0.630	−0.15 *	0.012	−0.68	0.120	−0.28 *	0.020	−0.54	0.150	−0.34 *	0.023
Education level	−0.13 *	0.039	−0.19 *	0.050	−0.04	0.267	−0.19	0.251	0.18 *	0.040	0.21 *	0.031
Occupation	0.01	0.494	0.01	0.544	0.20	0.967	0.01	0.824	0.04	0.986	0.03	0.947
Monthly income	0.14 **	0.004	0.12 *	0.016	0.01	0.564	0.01	0.369	0.19	0.256	0.04	0.990
Physical activity												
High	−0.13 *	0.050	−0.24 *	0.030	−0.44	0.965	−0.01 *	0.027	−0.30	0.360	−0.10	0.863
Moderate	−0.23 *	0.048	−0.34 *	0.050	−0.02	0.440	−0.03	0.382	−0.19	0.610	−0.11	0.875
Low	−0.02	0.309	−0.03	0.234	−0.04 *	0.017	−0.03	0.461	−0.28	0.463	−0.47	0.478
Dependent variables/Independent variables	POST-M (*n* = 31)											
	Calories				Dietary cholesterol Socioeconomic factors				Vitamin D			
Age	−0.04	0.586	−0.02	0.829	−0.01	0.193	−0.08	0.976	−0.51	0.301	−0.14	0.249
Education level	−0.16 *	0.038	−0.14 *	0.035	−0.01	0.331	−0.01	0.655	0.04	0.681	0.04	0.897
Occupation	0.01	0.508	0.03	0.436	0.01	0.402	0.12	0.789	0.03	0.589	0.11	0.562
Monthly income	0.14	0.204	0.03 *	0.027	0.11	0.975	0.02	0.498	0.04	0.941	0.01	0.477
Physical activity												
High	−0.04	0.832	−0.01	0.473	−0.22 *	0.019	−0.01	0.304	−0.09	0.134	−0.18	0.458
Moderate	−0.14 *	0.032	−0.02	0.439	−0.02	0.138	−0.03	0.359	0.41 **	0.002	0.22 *	0.011
Low	−0.06	0.800	−0.01	0.589	−0.02	0.163	−0.05	0.144	−0.07	0.952	−0.32	0.493

The significance level was accepted at * *p* ≤ 0.05; ** *p* ≤ 0.01.

## Data Availability

The data presented in this study are contained within the article.

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
