# Peer review of "The Effect of Ramadan Intermittent Fasting on Food Intake, Anthropometric Indices, and Metabolic Markers among Premenopausal and Postmenopausal Women: A Cross-Sectional Study"

_medicina, 2023, doi:10.3390/medicina59071191_

Round 1
Reviewer 1 Report
The current script (medicina-2361763) focused on the effect of Ramadan intermittent fasting on food intake, anthropometric Indices, and metabolic biomarkers among Pre- and Post-menopausal women. There are significant problems have to treat for reconsideration. These comments have to address in the revised version:
1.  Please check and revise the language and punctuation in Abstract.
2.  The novelty of this research should be inserted in the text clearly.
3.  The heading of Section 2, Methods and Methods?
4.  In Section 2.1, Study Participants, the participants were split into two groups: PRE-M women (21 - 43 years old) and POST-M women (42 - 68 years old). And in Abstract it was described “the study includes 62 women (31 PRE-M, 21-43 years, and 31 POST-M, 44-68 years)” which one is right? Is there any mistake in the following data table owing to the grouping?
5. The tables should be rearranged. Table 3 and 4 should be placed after Table 2.
6. In Table 4, there are 4 (12.9% ) person was found something wrong with Hypertension-I of blood pressure at R-1. So, are they healthy enough for the healthy group.
7. This question is not intended any offense, why could you take the following 21–30 days for taking the data of fasting (R2) at the end of Ramadan's third week instead of the end of the month.
Author Response
dear reviewee

Reviewer 2 Report
Dear Author,
I have revised your manuscript. I recommendation some points as below.
It should be explained the power analysis of the samples.
What is the reason of the second collection date? Why at the third week instead of the end?
Why this cut off point has been chosen for exclusion
How many hours of fasting occured due to the Ramdan fasting?Is it enough to see the potencial benefits of fasting according to literature?
Do the sleep habits of participants have been changed in Ramadan, indivudiuals may trend to change their sleeping habits as if they were living nocturnal.
It should be explained that even though the calorie change is statistically different, it is not significant as a percentage and that the positive results found are due to fasting not calorie restiction
In the discussion part, the changes in eating habits were put in the foreground rather than the fasting part. The results of fasting should be discussed more clearly.

Author Response
Dear reviewer

Reviewer 3 Report
Dear Editor
The age of PRE-M women was 21-43, and the age of POST-M women was 42-68. There is confusion about age. When a woman is 42 years old, is she post-menopausal or premenopausal? This uncertainty affects statistics and causes confusion. This issue needs to be reviewed.
‘’Both groups' BMI, waist circumference, waist-to-height ratio, percentage body fat, and fat mass were significantly lower at the end of Ramadan. After fasting, respondents’ triglycerides, total cholesterol, fasting blood glucose, and blood pressure (SBP, DBP) levels were significantly lower’’. Unfortunately, reaching such a conclusion is not correct to generalize. Because time is very limited.
RIF is a one-month process and is usually a process in which diet changes and often increases body weight. It is very important that the results of this study are reversed. However, the fact that the process is approximately one month is not suitable for generalization. Limitations of the subject should be given.
It would be useful to reconstruct the statistics and review the article again. It is useful to combine some tables.
With my compliment and best regards

Dear Authors
The age of PRE-M women was 21-43, and the age of POST-M women was 42-68. There is confusion about age. When a woman is 42 years old, is she post-menopausal or premenopausal? This uncertainty affects statistics and causes confusion. This issue needs to be reviewed.
‘’Both groups' BMI, waist circumference, waist-to-height ratio, percentage body fat, and fat mass were significantly lower at the end of Ramadan. After fasting, respondents’ triglycerides, total cholesterol, fasting blood glucose, and blood pressure (SBP, DBP) levels were significantly lower’’. Unfortunately, reaching such a conclusion is not correct to generalize. Because time is very limited.
RIF is a one-month process and is usually a process in which diet changes and often increases body weight. It is very important that the results of this study are reversed. However, the fact that the process is approximately one month is not suitable for generalization. Limitations of the subject should be given.
It would be useful to reconstruct the statistics and review the article again. It is useful to combine some tables.
With my compliment and best regards
Author Response
Dear reviewer

Reviewer 4 Report
I congratulate the authors for this research, nevertheless, there are some things that have to improve

Author Response
Dear reviewer

Round 2
Reviewer 3 Report
The authors made the desired corrections.